# Who Seeks Help from Live Chat Services? Demographic, Psychosocial, and Service Use Profiles of Young People Using a National Live Chat Mental Health Service

online counselling; synchronous chat; youth mental health; service engagement

**Corresponding author:**
Amanda Fitzgerald;
Email: amanda.fitzgerald@ucd.ie

Maria Tibbs[1,2] 🆔 and Amanda Fitzgerald[1]

[1]School of Psychology, University College Dublin, Ireland and [2]Psychology in Education Research Centre, Department of Education, University of York, UK

## Abstract

Online synchronous chat, or 'Live Chat', is distinguished by its real-time, anonymous, and text-based nature. There is limited understanding of the characteristics of those who choose Live Chat services compared to Blended Services. This study examined the demographic, psychosocial, and service engagement profiles of young people using the Irish Live Chat service, Jigsaw Live Chat, compared with those attending Jigsaw's Blended (in-person and/or video) support Service. Routine service-based data were analysed from 1,313 Live Chat and 3,604 Blended Service users. Reasons for attendance among Live Chat users seeking mental health support were analysed using content analysis. Live Chat users were more likely to be gender-diverse, older, and to report higher psychological distress than Blended Service users. Anxiety and low mood were common presenting issues. Attendance reasons varied, with over one-third citing multiple issues and many experiencing persistent distress. A minority attended for information or while waiting for other support. Overall, users reported high satisfaction. Live Chat users waited an average of 2.5 minutes, compared with typical waits of 1–2 months for the Blended Service. These findings highlight Live Chat as a distinct and essential access point for highly distressed and underserved youth, particularly those with minority gender and sexual identities.

## Impact statement

This study provides a timely examination of a commonly used but under-researched form of youth mental health support: online synchronous chat, or 'Live Chat'. It draws on national service data from Jigsaw Live Chat, an Irish single-session online chat service, providing a detailed exploration of the profile of young people choosing this modality. This project also compares some of these features with those of participants engaging in Jigsaw Brief Intervention, a Blended Service offering six to eight sessions of goal-focused mental health support. Young people accessing Live Chat waited an average of just 2.5 minutes to connect with a clinician, compared with typical wait times of 1–2 months for the Blended Service. Findings showed that Live Chat users were more likely to be young adults and gender diverse, and less likely to be male, than those accessing Blended Service. Across all age groups, users of the chat service reported significantly higher levels of psychological distress. Presentations were often complex, involving multiple and co-occurring difficulties, prolonged distress and concurrent use of other services. Overall, the findings show that Live Chat is not a substitute for in-person care but a distinct and essential access point, particularly for older adolescents and emerging adults who identify as gender diverse, LGBTQ+ or are experiencing high levels of psychological distress.



## Introduction

Mental health is a leading cause of disability among youth, with many difficulties emerging in adolescence and emerging adulthood (Gore et al., 2011; Vigo et al., 2016). Despite this, young people are one of the least likely groups to seek help for their mental health difficulties (Salaheddin and Mason, 2016) due to a range of structural barriers, such as cost, lengthy waiting times and complex referral procedures, that contribute to significant delays in receiving appropriate support. In addition to structural barriers, perceived stigma, embarrassment and negative beliefs about mental health services and professionals have consistently been shown to negatively impact help-seeking among youth (Gulliver et al., 2010; Velasco et al., 2020).

Online synchronous chat-based mental health support, often referred to as 'Live Chat', uniquely addresses help-seeking barriers by providing increased privacy and anonymity, which may reduce stigma (Navarro et al., 2019). Typically, these services operate on a drop-in basis, with minimal waiting times (Rickwood et al., 2019), and unlike in more traditional in-person or video-based supports, young people using chat modalities can view, appraise, refine and change the content of their messages, providing greater control over their self-presentation (Cheung et al., 2020). Existing international evaluations of synchronous chat services, such as Krisenchat (Germany), Kids Help Phone (Canada) and eHeadspace (Australia), indicate that they predominantly attract girls in mid-to-late adolescence seeking immediate, anonymous support for psychological distress and sensitive issues (Haner and Pepler, 2016; Rickwood et al., 2016; Eckert et al., 2022; Kohls et al., 2022).

### Jigsaw Live Chat: An Irish synchronous chat support service

Jigsaw – the National Centre for Youth Mental Health's Brief Intervention Service offers six to eight blended (in-person and/or video) sessions of free goal-focused mental health support to young people aged 12–25 years experiencing mild to moderate mental health difficulties across 13 sites in Ireland (O'Reilly et al., 2022). Jigsaw launched their online synchronous chat-based service 'Jigsaw Live Chat' in 2020, extending service reach across the Republic of Ireland. Jigsaw Live Chat provides anonymous, one-to-one, real-time, single-session mental health support delivered by clinicians via online chat.

### Evidence gaps and the current study

The recent global pandemic has further accelerated demand for mental health support. For instance, Kids Helpline reported that demand for their synchronous chat service increased during heightened coronavirus disease 2019 (COVID-19) lockdowns (Batchelor et al., 2021). Despite the increase in use of Live Chat services, little is known about the characteristics of young people choosing this modality and their motivations for doing so. To date, only two studies have detailed the characteristics of young people engaging with these Live Chat services. As a result of the growing interest in these technologies following the global pandemic, more research is needed to understand better why young people are choosing to seek support through chat-based modalities.

The current study aims to provide a demographic, psychosocial, and service engagement profile of young people attending Jigsaw Live Chat. To further investigate whether a unique cohort of young people present to Live Chat services, this study aimed to compare the profile of young people attending Jigsaw Live Chat to those who attended the Blended in-person and/or video Service. Specifically, the research questions are:

1. What is the demographic profile of young people accessing Jigsaw Live Chat?
2. What are the mental health difficulties and presenting levels of distress for young people accessing Jigsaw Live Chat?
3. How do young people engage with Jigsaw Live Chat?
4. Are young people satisfied with Jigsaw Live Chat?
5. How do the demographic profile and levels of psychological distress in Jigsaw Live Chat compare to those attending Jigsaw's Blended in-person and/or video Service?

## Methods

### Design

This study employed an exploratory retrospective design, analysing routine data from Jigsaw's Live Chat and Brief Intervention services (July 2020–December 2021) as part of their minimum datasets, that is, clinical and demographic data collected for the purposes of internal monitoring and evaluation.

### Interventions

*Jigsaw Live Chat* (henceforth referred to as Live Chat) is a one-to-one Live Chat service that provides free, confidential single-session mental health support to young people, available to young people Monday–Friday (10 am–5 pm/8 pm). Users register for the service using their email address via a purpose-built web-based platform, Dynamic Health. At registration, young people are prompted to enter information relating to their age, gender identity, sexuality, previous and current help-seeking information, and where they heard about the service. Users can choose to adopt a pseudonym or username for chat sessions to provide anonymity. Anonymised IDs are then used to link service user data. To engage in a one-to-one synchronous chat session, young people log onto the chat portal and enter a waiting queue pending clinician availability, at which point they are prompted to complete measures of psychological distress (Young People's Clinical Outcomes in Routine Evaluation (YP-CORE) and Clinical Outcomes in Routine Evaluation- 10 (CORE-10)) and their reasons for attendance. Once a clinician is available, the young person and clinician engage in a supportive chat session for ~45–50 min. Following the chat session, young people are invited to report their satisfaction with the service. At this point, clinicians, as part of their case notes, report the young person's presenting issues and whether they presented with risk behaviours (i.e., suicidal ideation, self-harm, a risk to others, and child protection concerns). Risk is managed through an iterative, staged escalation process, handled collaboratively and transparently with the young person. Situations indicating potential or imminent risk are escalated to the duty senior clinician or clinical manager, and emergency services are contacted if sufficient identifying information is provided.

*Jigsaw's Brief Intervention service* (henceforth referred to as the Blended Service) offers brief, goal-focused therapeutic support to young people, Monday–Friday (9 am–5.30 pm/7.30 pm). Once referred, young people are invited to attend an initial screening assessment, within which measures of psychological distress (YP-CORE and CORE-10) are administered by the clinician. This is followed by a more intensive screening session, where, in line with the Power Threat Meaning Framework (Johnstone et al., 2018), information relating to the young person's psychosocial and demographic profile and the contextual factors that may be contributing to or perpetuating their mental health difficulties are recorded by the clinician. After screening sessions, young people may be invited to participate in approximately six to eight therapeutic intervention sessions, each lasting around 1 hour. Wait times for these services vary nationally, with the median wait time in 2020 reported as ~63 days between referral and the first appointment. Before the onset of the global pandemic in March 2020, Jigsaw's Brief Interventions were delivered in person by a clinician in one of their $N = 13$ sites in 10 counties across Ireland (Cork, Donegal, Dublin [four sites], Galway, Kerry, Laois/Offaly, Limerick, Meath, Tipperary and Wicklow). However, in response to the pandemic and the changing needs of young people, Jigsaw began to offer a Blended model of

support in 2020, including video, phone and in-person (O'Reilly et al., 2022).

Clinicians across both services are trained in disciplines like psychology and occupational therapy, and use therapeutic approaches such as solution-focused, cognitive behavioural, acceptance and commitment and compassion-focused therapies (O'Reilly et al., 2022). For further detail on the pathways to support across both services, see Figure 1.

### Participants and inclusion criteria

#### Live chat service (Jigsaw Live Chat)
Of the 4,013 young people (aged 12–25 years) registered with the Live Chat service, 32.9% ($n = 1,313$) engaged in at least one chat exchange, while 5.85% ($n = 240$) left the queue before their session. Among those engaging in a chat session, 92.9% ($n = 1,220$) sought mental health support, 3.5% ($n = 46$) did not report their reason, 2.5% ($n = 33$) supported another, and 1.1% ($n = 14$) sought information. Demographic and engagement data are presented for all attendees, but psychological distress and presenting issues are reported for those seeking mental health support ($n = 1,220$). Gender responses marked as 'prefer not to say' ($n = 22$) were removed from inferential analyses.

#### Blended Service (Jigsaw Brief Intervention)
From 3,831 referrals, individuals aged below 12 or over 25 years ($n = 41$) were excluded. Since CORE measures are age-specific (YP-CORE: 12–16 years; CORE-10: 17+ years), cases with measures that did not meet these age criteria (YP-CORE: $n = 8$; CORE-10:

$n = 87$) or those with incomplete measures ($n = 91$) were also removed, leaving a final sample of 3,604. Gender comparisons were only possible for the final 6 months (June–December) of the study due to changes in data collection (adopting measurements of gender diversity) in the Blended Service.

### Measures
Data were part of Jigsaw's routinely collected minimum datasets for the Live Chat and Blended services and included demographic, psychosocial and service engagement characteristics of young people using the services.

#### Demographic characteristics
Demographic data were captured at registration for the Live Chat service and at referral for the Blended Service. Data included age, gender, sexuality, ethnicity and county of residence. For the Live Chat service, this information was self-reported, while for the Blended Service, it was recorded by clinicians.

#### Psychosocial characteristics
Psychological distress was measured using the YP-CORE (12–16 years; Twigg et al., 2016) and the CORE-10 (17+ years; Barkham et al., 2013). Scores range from 0 to 40, with higher scores indicating greater psychological distress. Clinical cut-offs for the YP-CORE were applied using age- and gender-specific norms from the measure's manual (see Supplementary Material). For the CORE-10, a score above 11 indicated clinically significant distress. While gender differences may be explored descriptively, no separate gender-

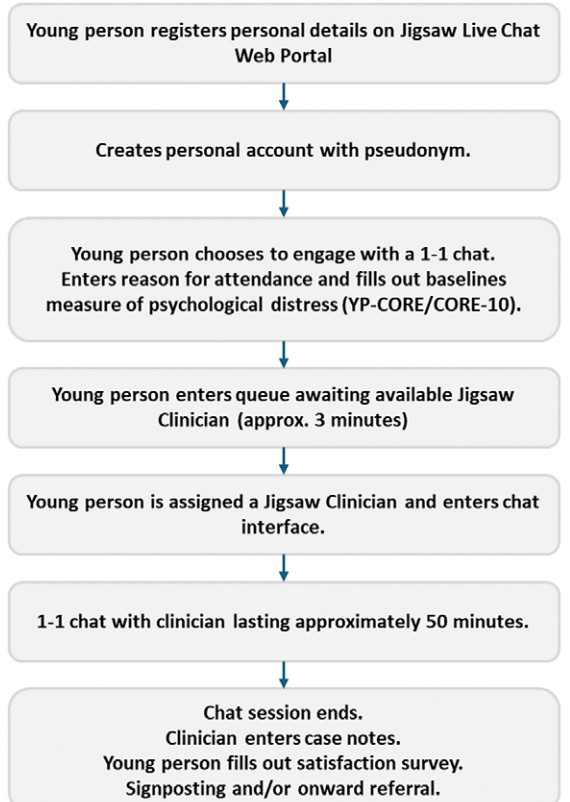

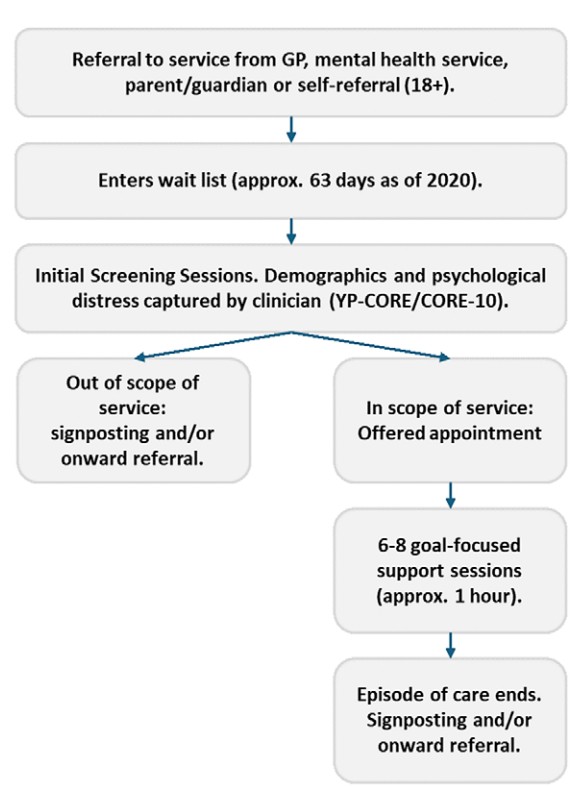

**Figure 1.** Pathways for Accessing the Live Chat and Blended Services.

based cut-offs are recommended. No manual-validated clinical cut-off exists for gender-diverse youth on either measure. McDonald's Omega for YP-CORE was $\omega = .75$ (Live Chat) and $\omega = .80$ (Blended Service); for CORE-10, $\omega = .75$ (Live Chat) and $\omega = .74$ (Blended Service). Clinicians retrospectively reported presenting issues following each Live Chat session, selecting from 16 predefined categories reflecting 'self' or 'contextual' issues. Young people provided qualitative responses on their reasons for attending prior to their first session, after registering and completing demographic information; at the point of initiating a chat, they completed either the YP-CORE or CORE-10 and restated their reason for attendance.

### Service engagement characteristics

Service engagement data assessed the ways the young person interacted with the Live Chat service. This included data self-reported by the young person, such as where the young person heard about Jigsaw, current and previous service use and service satisfaction. Service satisfaction was captured using a modified version of the eheadspace satisfaction questionnaire (Rickwood et al., 2019). This brief measure has nine items, scored on a 5-point Likert, with a range of 9–45. McDonald's omega for the current study was $\omega = .91$. Additional engagement metrics (wait time, session duration, latency and number of textual exchanges) were automatically captured by the Live Chat service's data management system. See Supplementary Materials for definitions of these key engagement metrics.

### Data analysis

Data were analysed with SPSS (v29.0). Age and gender differences in Live Chat data were assessed using t-tests, analysis of variances (ANOVAs) and Chi-square tests. Distress levels were analysed separately for age groups defined by the YP-CORE (12–16 years) and CORE-10 (17+ years), while broader age categories (12–14, 15–17 and 18–25 years) were used for developmental comparisons. A conservative p-value (< .01) was used due to exploratory analyses and unequal cell sizes. To avoid making many low-powered comparisons across multiple demographic variables, exploratory analyses were restricted to age and gender. For the same reason, only the most prevalent presenting issues and risk behaviours (suicidal ideation and self-harm) were examined, as sparse data in other categories may have produced unstable estimates and increased the likelihood of type I error.

### Inductive content analysis of open-ended reasons for attendance and development of coding framework

Reasons for attendance were analysed using a combination of inductive and deductive content analysis, following the approach outlined by Crowe et al. (2015) and Graneheim and Lundman (2004). This enabled a flexible and iterative process suited to capturing both emergent and evidence-informed themes. Initially, the data underwent inductive open coding, during which descriptive units of text were extracted directly from participants' stated reasons for attendance. Segments capturing key concepts were identified and labelled using NVivo software. These descriptive units were then grouped into broader categories and thematic areas (see Supplementary Materials for a worked example). A coding framework was developed and refined iteratively as new data emerged. To ensure reliability, 20% of responses (n = 250) were double-coded in Microsoft Excel by a second coder (AF), yielding

high inter-rater reliability ($\kappa = .96$). Frequencies of overarching categories and thematic areas were calculated, accounting for instances where multiple reasons were reported. Cases indicating prolonged difficulties were coded as 'chronic or persistent', while those describing escalating severity were coded as 'increasing or worsening'.

### Ethical approval

This study received ethical approval from the Jigsaw Research Ethics Committee (JREC-E-2020-001) and subsequently received additional approval from the Human Research Ethics Committee, University College Dublin (HS-E-21-132-Tibbs-Fitzgerald).

## Results

### Demographic characteristics

The majority of those who successfully engaged in their first chat session were female (72.7%; n = 954), heterosexual (63%; n = 827) and White Irish (80.3%; n = 1,054). The mean age of presentation at an initial chat session was 18 years of age (standard deviation [SD] = 3.23). Among the young people who reported their gender as 'other – please specify' (2.1%; n = 27), the majority identified as non-binary or demigender (65.35%; n = 17). The top counties of residence (Dublin, Cork, Galway, Meath and Limerick) largely align with the most populated counties in Ireland (Dublin, Cork, Galway, Kildare and Meath; Central Statistics Office, 2022), and with the Blended Services' largest number of referrals during the same period (July 2020 to end December 2021; Dublin, Cork, Galway, Limerick and Laois/Offaly). No significant differences in demographic characteristics between those young people who registered for the service but did not engage in a chat session and those who engaged in a chat session were observed. For a detailed description of the demographic characteristics of those who registered and those who attended a chat session, see Table 1.

### Comparison of demographics between the Live Chat and Blended Services

Upon removal of those who selected prefer not to say, participants in the Blended Service comprised 69.01% females (n = 2,487), 30.66% males (n = 1,105), 0.19% identifying as another gender (e.g., non-binary; n = 7), and 0.14% who were not sure or questioning (n = 5). In comparison, attendees of the Live Chat service comprised 73.20% females (n = 954), 21.25% males (n = 277), 2.07% identifying as another gender (n = 27), and 2.53% who were not sure or questioning (n = 33).

A chi-square test indicated significant associations between gender and the service, $\chi^2 = 34.14$, $df = 2$, $p < .001$. Fewer males (z = -2.2) and more gender-diverse young people (z = 4.7) presented to Jigsaw's Live Chat service compared with the Blended Service. Young people attending Live Chat were older (M = 17.99; SD = 3.23) than those attending the Blended Service (M = 15.93, SD = 3.0), $t(4915) = 20.18$, $p < .001$, $d = .67$.

### Psychological distress in the Live Chat Service

Of the N = 1,313 young people who received a chat session, 92.9% (n = 1,220) were seeking mental health support. Those aged 12–16 years

**Table 1.** Demographic characteristics of young people registering or attending a Live Chat session (N = 4,013)

| | Registered (N = 2,700) | | Attended (N = 1,313) | |
|---|---|---|---|---|
| | n (%) | M (SD) | n (%) | M (SD) |
| **Gender** | | | | |
| Female | 1,972 (73.04) | | 954 (72.66) | – |
| Male | 553 (20.48) | | 277 (21.10) | – |
| Other | 75 (2.78) | | 27 (2.06) | – |
| Not sure/ questioning | 62 (2.30) | | 33 (2.51) | – |
| Prefer not to say | 38 (1.41) | | 22 (1.68) | – |
| **Age group (years)** | | | | |
| 12–14 | 515 (19.07) | | 207 (15.77) | – |
| 15–17 | 882 (32.67) | | 404 (30.77) | – |
| 18–20 | 713 (26.41) | | 411 (31.30) | – |
| 21–25 | 590 (21.85) | | 291 (22.16) | – |
| **Age** | – | 17.67 (3.30) | – | 17.99 (3.23) |
| **Sexual identity** | | | | |
| Heterosexual | 1,612 (59.70) | | 827 (62.99) | – |
| Gay | 59 (2.19) | | 28 (2.13) | – |
| Lesbian | 108 (4.00) | | 41 (3.12) | – |
| Bisexual | 432 (16.00) | | 196 (14.93) | – |
| Other | 59 (2.19) | | 40 (3.05) | – |
| Not sure/ questioning | 274 (10.15) | | 88 (6.70) | – |
| Prefer not to say | 126 (4.67) | | 93 (7.08) | – |
| **Ethnicity** | | | | – |
| White Irish | 2,114 (78.30) | | 1,054 (80.27) | – |
| Any other White background | 204 (7.56) | | 95 (7.24) | – |
| Asian or Asian Irish | 102 (3.78) | | 44 (3.35) | – |
| Black or Black Irish | 92 (3.41) | | 33 (2.51) | – |
| Background | 82 (3.04) | | 42 (3.20) | – |
| Irish Traveller | 14 (.52) | | 3 (.23) | – |
| Roma | 1 (.04) | | 1 (.08) | – |
| Other | 42 (1.56) | | 12 (.91) | – |
| Prefer not to say | 49 (1.81) | | 29 (2.21) | |
| Missing | | | | |
| **County (top 5)** | | | | |
| Dublin | 972 (36.00) | | 443 (33.74) | – |
| Cork | 301 (11.15) | | 164 (12.49) | – |
| Galway | 170 (6.30) | | 107 (8.15) | – |
| Meath | 142 (5.26) | | 97 (7.39) | – |
| Limerick | 127 (4.70) | | 53 (4.04) | – |

(N = 427) had an average YP-CORE score of 25.96 (SD = 6.42), above the midpoint of 20. Those aged 17–25 years (N = 793) had an average CORE-10 score of 22.9 (SD = 6.30), above the midpoint of 20.

Two 3 × 2 ANOVAs examined age and gender differences in YP-CORE and CORE-10 scores. A small but significant main effect of gender was found on YP-CORE scores (those aged 12–16 years), $F(2, 413) = 8.31$, $p < .001$, $\eta p^2 = .04$. Post hoc Tukey tests showed that gender-diverse individuals (M = 29.43, SD = 5.14) reported higher distress than males (M = 24.27, SD = 6.43) and females (M = 25.92, SD = 6.39). No main or interaction effects for age and gender were found across CORE-10 psychological distress scores.

Independent *t*-tests examined differences in distress by sexuality. LGBTQ+ individuals (M = 27.75, SD = 5.83) reported higher YP-CORE distress scores than heterosexual individuals (M = 24.91, SD = 6.60), $t(392) = -4.38$, $p < .001$, $d = .42$. However, no differences in distress were observed on CORE-10 scores across sexuality.

### Clinician-reported presenting issues and risk presentation to the Live Chat Service

Following the chat session, clinicians reported anxiety as the most prevalent mental health difficulty presented by young people (60.4%; N = 737), followed by mood difficulties, and social difficulties (43.4%; N = 529). Notably, clinicians reported that out of all N = 1,220 chat sessions, at least one risk behaviour was present in 34.8% (N = 425) of the sample. Risk, as defined by the service, is the expression of suicidal ideation or tendencies, self-harm behaviours, issues relating to child protection or expression of risk to others. For full descriptions of presenting issues and risk behaviours, see Table 2.

Chi-square analyses examined associations between gender, the top two risk behaviours (suicidal ideation and self-harm) and the top five presenting issues (anxiety, low mood, difficulties at school and problems at home), while independent *t*-tests assessed age differences. See Table 3 for detailed results.

### Content analysis of reasons for youth attendance at the Live Chat Service

Of 1,313 Live Chat attendees, 1,258 provided qualitative responses on their reasons for attendance, totalling 2,010 responses. These were categorised into 'Mental Health Difficulties' (81.1%), 'Support Requests' (14.5%), 'Information Requests' (3.1%) and 'Other' (1.3%). For further details on the breakdown of these thematic areas, see Supplementary Material.

Of the reported mental health difficulties, $n = 126$ (7.8%) were chronic or persistent, with young people describing ongoing struggles (e.g., '*I have been feeling down for several months*'). Additionally, $n = 85$ (5.25%) reported increases in severity of symptoms before their Live Chat attendance (e.g., '*My anxiety has been worse than ever, my chest physically hurts*'). Over a third of those attending for a mental health concern reported multiple issues at the time of logging into Live Chat ($n = 450$; 35.8%). The most common co-occurring difficulties were anxiety and low mood ($n = 67$), anxiety and general distress ($n = 22$) and anxiety and stress ($n = 20$).

### Comparison with the Blended Service

Young people aged 12–16 years attending the Live Chat service reported higher psychological distress on the YP-CORE (M = 25.97, SD = 6.42) compared to those attending the Blended Service (M = 18.08, SD = 7.11), $t(2683) = 21.32$, $p < .001$, $d = 1.13$. Similarly, those aged 17–25 years using Jigsaw's Live Chat had higher CORE-10 scores (M = 22.90, SD = 6.31) than their Blended Service counterparts (M = 17.28, SD = 6.01), $t(2080) = -20.32$, $p < .001$, $d = .92$.

CORE risk item analysis showed that 12- to 16-year-olds in Live Chat scored higher on '*I've thought of hurting myself*' (M = 1.41, SD = 1.41) compared to Blended Service attendees (M = .43, SD = .85),

**Table 2.** Live Chat service YP-CORE clinical ranges and clinician-reported presenting issues and risk behaviours

| Presenting issues (N = 1,220) | n (%) |
|---|---|
| **YP-CORE clinical range** | |
| Males aged 12–13 years | 19 (100) |
| Males 14–16 years | 58 (95.1) |
| Females 12–13 years | 69 (92.0) |
| Females 14–16 years | 231 (91.1) |
| **CORE–10 clinical range** | |
| Females | 600 (95.7) |
| Males | 187 (93.5) |
| **Presenting issues** | |
| Anxiety | 737 (60.4) |
| Mood difficulties | 508 (41.6) |
| Social difficulties | 529 (43.4) |
| Difficulties at school | 408 (33.4) |
| Problems at home | 400 (32.8) |
| Suicidal behaviours | 231 (18.9) |
| Other | 248 (20.3) |
| Unknown | 129 (10.6) |
| Low self-esteem | 110 (9.0) |
| Loneliness | 105 (8.6) |
| Eating | 66 (5.4) |
| Sleep | 65 (5.3) |
| Anger | 51 (4.2) |
| Trauma | 37 (3.0) |
| Identity issues | 34 (2.8) |
| **Risk behaviours** | |
| All risk | 425 (34.8) |
| Suicidal ideation | 263 (21.6) |
| Self-harm | 234 (19.2) |
| Child protection | 8 (0.7) |
| Child welfare | 8 (0.7) |
| Other | 9 (0.7) |
| Risk to others | 7 (0.6) |
| Concern about another | 2 (0.2) |

*Note*: More than one presenting issue could be reported. The YP-CORE is completed by 12- to 16-year-olds and the CORE-10 by 17- to 25 -year-olds. YP-CORE clinical range rows follow age- and gender-specific norms; CORE-10 uses a single cut-off across gender and age. A cut-off to denote the clinical range of distress has not yet been determined for gender-diverse populations.

$t(2721) = 19.46$, $p < .001$, $d = 1.03$. Similarly, 17+ participants in Live Chat scored higher on '*I have made plans to end my life*' ($M = .44$, $SD = .91$) than Blended attendees ($M = .08$, $SD = .33$), $t(2096) = 12.66$, $p < .001$, $d = .57$.

A higher proportion of Live Chat users met clinical distress criteria compared to Blended users across all age and gender groups. Distress was highest among males aged 12–13 years

(100%) and females aged 17+ years (95.7%) in the Live Chat service. Full chi-square comparisons are presented in Table 4.

### Service engagement characteristics

Young people accessing the Live Chat Service ($N = 1,313$) averaged 1.67 chat sessions ($SD = 2.01$), with 25.6% ($n = 337$) engaging in more than one session. Of those engaging in more than one session, most had 2–4 chats (21.4%; $n = 282$), while 3.1% ($n = 41$) had 5–9 and 1.1% ($n = 14$) had over 10 chats. The average wait time for a successful session was 2.54 minutes ($SD = 1.84$), compared to 5.86 min ($SD = 15.10$) for those who quit the queue ($N = 240$). Initial chat sessions averaged 29.92 message exchanges ($SD = 14.61$), with young people sending ~16.27 messages ($SD = 10.15$) and clinicians sending 13.65 messages ($SD = 5.86$). The average duration of a chat session was 52.84 minutes, although variation was considerable ($SD = 19.27$). Response latency averaged 3.75 min ($SD = 1.92$) for young people and 4.51 min ($SD = 1.82$) for clinicians.

ANOVA results showed age had a small effect on session attendance, $F(3, 1309) = 38.35$, $p < .001$, $\eta p^2 = .08$, with those aged 12–14 years ($M = 1.92$, $SD = 2.76$) and 15–17 years ($M = 1.78$, $SD = 2.48$) engaging in more chat sessions than those aged 21–25 years ($M = 1.31$, $SD = 1.10$). No differences were found by gender or psychological distress profiles.

### Previous and current mental health service use

Over half of users (56.2%; $n = 739$) had accessed other mental health services, including 10.3% ($n = 256$) who had attended the Blended Service previously. Additionally, 28.4% ($n = 374$) were currently engaged in other services, with 6.4% ($n = 84$) attending the Blended Service.

### Where young people heard about the live chat service

Most users learned about live chat via Google (28.6%; $n = 378$), social media (19.1%; $n = 252$), family (10.4%; $n = 137$) or peers (10.1%; $n = 133$). Other sources included doctors (7.2%; $n = 95$), schools (7.0%; $n = 93$), and helplines (1.5%; $n = 20$).

### Satisfaction with the Live Chat service

Survey response rates ranged from 33% to 40.5%, with an average satisfaction score of 39.45 ($SD = 6.43$). Most young people felt listened to (97.4%; $n = 518$), spoke about desired topics (91.5%; $n = 487$) and discussed skills or resources (87%; $n = 463$). Many reported better understanding their problem (78%; $n = 415$), feeling more hopeful (69.9%; $n = 372$), and coping better (66.9%; $n = 313$). Others felt that sessions improved their ability to manage their problems (57.4%; $n = 252$), general functioning (56.6%; $n = 251$), or relationships (43.7%; $n = 189$).

## Discussion

Live Chat users were mostly female, White and heterosexual, with an average age of 18 years. Compared with Blended Service users, they were more likely to be gender diverse, older and to report higher psychological distress. The primary presenting issues were anxiety, low mood and social and relationship difficulties. When asked why service users were attending the Live Chat service, young people often reported needing both mental health support and informational support. On average, users attended 1.67 sessions, with one-quarter returning for additional sessions. Service satisfaction was high, and the average wait time was 2.54 minutes, with clinicians taking

**Table 3.** Clinician-reported presenting issues and risk behaviours to the Live Chat service across age and gender

| | Female (n = 893) | | | Male (n = 257) | | | Gender diverse (n = 53) | | | Total sample (N = 1,203) | | |
|---|---|---|---|---|---|---|---|---|---|---|---|---|
| | n | % | $Z^{adjusted}$ | n | % | $Z^{adjusted}$ | n | % | $Z^{adjusted}$ | $\chi^2$ | Age M (SD) | t |
| **Anxiety** | | | | | | | | | | 8.59* | | −1.66 |
| Yes | 563 | 63.0 | 2.8* | 136 | 52.9 | −2.9* | 32 | 60.4 | −0.1 | | 18.11 (3.24) | |
| No | 330 | 37.0 | −2.8* | 121 | 47.1 | 2.9* | 21 | 39.6 | 0.1 | | 17.8 (3.24) | |
| **Low mood** | | | | | | | | | | 1.48 | | −.64 |
| Yes | 365 | 40.9 | −.8 | 109 | 42.4 | .3 | 26 | 49.1 | 1.1 | | 18.06 (3.22) | |
| No | 528 | 59.1 | .8 | 148 | 57.6 | −.3 | 27 | 50.9 | −1.1 | | 17.94 (3.26) | |
| **Social difficulties** | | | | | | | | | | 3.87 | | −1.26 |
| Yes | 381 | 42.7 | −1.1 | 124 | 48.2 | 1.7 | 19 | 35.8 | −1.2 | | 18.12 (3.30) | |
| No | 512 | 57.3 | 1.1 | 133 | 51.8 | −1.7 | 34 | 64.2 | 1.2 | | 17.88 (3.12) | |
| **Difficulties at school** | | | | | | | | | | 5.61 | | 1.84 |
| Yes | 305 | 34.2 | .7 | 75 | 29.2 | −1.7 | 24 | 45.3 | 1.8 | | 17.75 (3.07) | |
| No | 588 | 65.8 | −.7 | 182 | 70.8 | 1.7 | 29 | 54.7 | −1.8 | | 18.11 (3.32) | |
| **Problems at home** | | | | | | | | | | 3.40 | | 2.15 |
| Yes | 291 | 32.6 | .0 | 78 | 30.4 | −.9 | 23 | 43.4 | 1.7 | | 17.70 (3.30) | |
| No | 602 | 67.4 | .0 | 179 | 69.6 | .9 | 30 | 56.6 | −1.7 | | 18.12 (3.23) | |
| **Any risk behaviour** | | | | | | | | | | 24.30* | | 9.87* |
| Yes | 309 | 34.6 | −.1 | 74 | 28.8 | −2.2* | 34 | 64.2 | 4.6* | | 16.78 (6.06) | |
| No | 584 | 65.4 | .1 | 183 | 71.2 | 2.2* | 19 | 35.8 | −4.6* | | 18.63 (3.15) | |
| **Suicidal ideation** | | | | | | | | | | .29 | | 2.02 |
| Yes | 188 | 21.1 | −.4 | 58 | 22.6 | .5 | 11 | 20.8 | −.1 | | 17.63 (3.0) | |
| No | 705 | 78.9 | .4 | 199 | 77.4 | −.5 | 42 | 79.2 | .1 | | 18.08 (3.30) | |
| **Self-harm** | | | | | | | | | | 58.41* | | 14.03* |
| Yes | 175 | 19.6 | .8 | 25 | 9.7 | −4.3* | 29 | 54.7 | 6.8* | | 15.50(2.50) | |
| No | 718 | 80.4 | −.8 | 232 | 90.3 | 4.3* | 24 | 45.3 | −6.8* | | 18.57(3.12) | |

*Note*: * Indicates statistical significance at *p* <.01 level.

**Table 4.** Clinical caseness across Live Chat and in-person or video services

| Age and gender | n (Live chat) | % | $Z^{adjusted}$ | n (Blended) | % | $Z^{adjusted}$ | $\chi^2$ |
|---|---|---|---|---|---|---|---|
| 12–13 years females | 69 | 92.00 | +4.70** | 372 | 65.40 | −1.70 | 20.51 |
| 14–16 years females | 231 | 91.10 | +7.07** | 740 | 69.30 | −3.45** | 48.86 |
| 12–13 years males | 19 | 100.00 | +3.02* | 196 | 66.90 | −0.77 | 7.65 |
| 14–16 years males | 58 | 95.10 | +5.56** | 243 | 58.30 | −2.13* | 29.35 |
| 17+ years females | 600 | 95.70 | +5.86** | 760 | 86.70 | −4.95** | 33.25 |
| 17+ years males | 187 | 93.50 | +3.71** | 337 | 82.40 | −2.60* | 12.87 |

*Note*: *p < .01, **p < .001. The YP-CORE is completed by 12- to 16-year-olds and the CORE-10 by 17- to 25-year-olds. A cut-off to denote the clinical range of distress has not yet been determined for gender-diverse populations using these measures.

slightly longer to respond than users. Collectively, these patterns suggest that Live Chat is operating as an immediate, low-barrier point of support for young people who experience significant psychological distress, varied presenting issues, and a combination of emotional and informational needs.

Consistent with prior research, findings showed a predominance of females accessing online chat services (Haner and Pepler, 2016; Rickwood et al., 2016; Eckert et al., 2022). This also reflects more general trends in help-seeking, whereby females demonstrate an increased tendency to seek support and typically present with higher overall levels of psychological distress (Clark et al., 2018; Rafal et al., 2018; Haavik et al., 2019; Campbell et al., 2021). Still, the chat-based modality may be further influencing these patterns, where females are more likely to engage in text-based communication than males (Kimbrough et al., 2013; Rosenfeld et al., 2018). Women also tend to adopt more emotive textual communication

styles, including more frequent emoji use (Rungta, 2015; Koch et al., 2022), whereas males typically favour shorter and more concise messages (Rosenfeld et al., 2018). This preference may partly explain lower engagement among males with chat-based platforms that require more detailed and emotive textual communication.

Gender differences were also observed across presenting issues and psychological distress. Consistent with reported prevalence rates, females were more likely to present with anxiety (McLean et al., 2011; Dooley et al., 2019). Gender-diverse youth, on the other hand, were more likely to present with self-harm (White et al., 2023). Additionally, gender-diverse individuals aged 12–16 years exhibited significantly higher distress levels than both males and females. This finding aligns with evidence suggesting that early to mid-adolescence is a period for heightened gender questioning and intensified pressure to conform to gender norms, both of which are associated with greater mental health challenges (Verschueren et al., 2017; King et al., 2019). Overall, levels of psychological distress were comparable to those reported in other online synchronous chat-based services, with previous research indicating that chat users typically present with higher distress than those accessing in-person interventions (Rickwood et al., 2016). Similarly, rates of risk presentations, including suicidal ideation and self-harm, were consistent with findings from other evaluations of synchronous chat-based services (Haner and Pepler, 2016; Kohls et al., 2022).

The peak age of presentation to Jigsaw's Live Chat was 18 years, older than previously reported by youth-focused chat services (Haner and Pepler, 2016; Rickwood et al., 2019; Batchelor et al., 2021; Eckert et al., 2022). Despite this, younger cohorts were more likely to present with self-harm and displayed higher distress than older users. Still, younger individuals were less likely to use Live Chat despite the absence of the need for parental consent to attend. This may reflect the tendency for young adults (18–25 years) to self-initiate help-seeking (Wood et al., 2018), while younger adolescents are more influenced by parental figures (Rickwood et al., 2015). This is reflected in the top ways in which users reported hearing about the service (Google, friends and social media).

Qualitative findings provide further insight into the complex presentation of those attending Jigsaw's Live Chat service. Findings revealed that many young people attended the service for multiple, and sometimes worsening, reasons. Previous research has indicated that young people attending chat-based services often have high outcome expectancies (Dowling and Rickwood, 2016). Given that many Live Chat services are designed as single-session interventions, these elevated expectations, alongside complex and high-distress presentations, present important challenges for outcome assessment. Evidence from the in-person single session therapy literature highlights the role of expectation management, as well as clear goal and focus setting, in supporting effective outcomes (Hoyt and Talmon, 2015; Dryden, 2020). These components may be even more crucial in the context of Live Chat, where time-limited support must also respond to heterogeneous and often acute mental health needs.

More than half of Live Chat users had previously accessed a mental health service. Of these, almost one-third of users (28.4%) reported using other services concurrently. This pattern suggests that Live Chat may function as a form of bridging or supplementary support within broader mental health care pathways. A smaller subset (4.5%) explicitly cited using Live Chat while on a waitlist or between appointments, reflecting the role of digital supports in addressing service delays (McGorry et al., 2022). Consistent with evidence that young people increasingly seek flexible, on-demand mental health support tailored to their needs (Tatlow-Golden and McElvaney, 2015; Holding et al., 2022), these findings suggest that

Live Chat meets this demand by providing immediate access to help. Immediacy appears central to the appeal of chat-based support. For young people experiencing acute or escalating distress, the ability to connect with a clinician within minutes may offer both a sense of agency and psychological containment. Still, further research is needed to understand how the immediacy and flexibility of online services influence engagement and continuity of care within traditional service pathways.

Aligning with its intended use, most young people attended just one session, with 25.6% returning for additional chat sessions. However, in the absence of follow-up outcome measurement, it remains unclear whether a single session adequately meets the needs of this highly distressed subgroup, particularly given the complexity of their presentations. Live chat users sent more messages and responded faster than clinicians, a pattern seen in other text-based services (Sindahl and van Dolen, 2020). It remains uncertain whether this reflects broader digital communication norms among young people or features unique to the therapeutic chat environment. Rather than a limitation, response latency may allow both clinicians and young people to compose more deliberate, reflective exchanges, which may be less feasible in face-to-face interactions.

Waiting experiences also appeared to shape engagement. Young people who exited the queue before their first chat session experienced longer wait times than those who engaged (5.86 vs. 2.54 minutes). Although these waits are minimal compared with traditional services, where the typical wait for Blended Support ranged from 1 to 2 months in 2020, the subjective experience of waiting may differ across modalities. Experiences of wait times within online synchronous chat services remain largely unexplored, perhaps reflecting the assumption that waits of only a few minutes are inconsequential when compared with traditional services. Yet, this assumption may overlook the unique expectations of immediacy in real-time digital support, where even brief delays can shape engagement and satisfaction. This phenomenon has been examined more extensively in the online chat-based customer service literature, where researchers show that individuals tend to overestimate short wait durations, which in turn reduces satisfaction (Hong et al., 2013; McLean and Wilson, 2016). Although preliminary, these findings highlight the need to consider wait times, even those lasting only a few minutes, as potential barriers to engagement.

Despite these challenges, young people were generally satisfied with Jigsaw's Live Chat service and commonly reported feeling listened to and understood. In contrast, fewer reported improvements in psychosocial outcomes, such as hope, coping or relationships. This likely reflects the wide range of reasons for attending and the varied nature of many presenting concerns, rendering broad, uniform outcome indicators poorly suited to a single-session chat service. These findings highlight the need for outcome assessment grounded in a clear programme theory, ensuring alignment between why young people attend, what the service can reasonably provide, and the types of change that can occur within a single session. Revising the outcome questions, for example, through the adoption of idiographic tools such as the Session Wants and Needs Outcome Measure (de Ossorno Garcia et al., 2021), or dynamic approaches such as machine learning could provide a more accurate picture of service impact.

## Strengths and limitations

This study had some limitations, including differences in the sample sizes across service samples and differences in data collection methods that hindered some demographic comparisons.

Additionally, data were collected during the COVID-19 lockdowns, which may have influenced distress levels. Nevertheless, the present study provides valuable insights into the demographic, psychosocial, and service engagement profiles of young people using Live Chat, highlighting both the complexity of their mental health needs and the role of anonymous real-time online services in contemporary help-seeking.

## Conclusion

This study highlights the distinct profiles of young people using an Irish Live Chat service. Compared with those seeking Blended video and/or in-person Support, Live Chat users were predominantly female, more likely to identify as gender diverse, older and reported higher psychological distress. Gender-diverse and LGBTQ+ young people, particularly those aged 12–16 years, showed the highest distress. Presentations commonly involved anxiety, low mood and social challenges, and many users had prior or concurrent service involvement, indicating substantial complexity. Although most engaged in a single session, more than one-quarter returned for additional support, reinforcing the level of need among some users. The present findings indicate that the single-session Live Chat model provides timely access and may ease pressure on traditional services, highlighting the importance of integrating live chat within a broader service pathway so that rapid access is matched with appropriate follow-on care when needed.

**Open peer review.** To view the open peer review materials for this article, please visit http://doi.org/10.1017/gmh.2025.10120.

**Supplementary material.** The supplementary material for this article can be found at http://doi.org/10.1017/gmh.2025.10120.

**Data availability statement.** Data are not publicly available due to the sensitive nature of the service-related data collected.

**Acknowledgements.** Authors would like to thank the young people who provided their data for the present study.

**Author contribution.** MT devised the study, extracted and analysed the data and wrote the manuscript. AF devised the study and conducted secondary coding of the qualitative data. All authors edited and approved the final manuscript.

**Financial support.** This research was funded by the Irish Research Council Employment Based Postgraduate Scheme (EBPPG/2020/91).

**Competing interests.** The authors declare none.

**Ethical standard.** This study received ethical approval from the Jigsaw Research Ethics Committee (JREC-E-2020-001) and subsequently received additional approval from the Human Research Ethics Committee, University College Dublin (HS-E-21-132-Tibbs-Fitzgerald).

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
