## [Reviewer Report]

This manuscript presents an important and timely contribution to the growing field of digital mental health, offering valuable insights into the distinct characteristics and needs of young people who engage with online synchronous chat counselling. By comparing users of Jigsaw’s Live Chat service with those accessing in-person and video-based interventions, the study advances understanding of how digital platforms can serve as critical access points for highly distressed and gender-diverse youth. The rigorous mixed-methods approach, combining quantitative profiling with qualitative content analysis, provides a rich and nuanced picture of user engagement and motivation. While the paper is both interesting and scientifically significant, several areas would benefit from further refinement and clarification to enhance the clarity, coherence, and overall impact of the work.

Introduction

Please expand the introduction by incorporating findings from other evaluations of synchronous chat-based support services (e.g., Eckert et al., 2022) conducted in different countries. In particular, consider referencing available evidence regarding user demographics and engagement characteristics, as these few existing studies can help contextualize and justify the research questions derived in this study.

Please enrich the description of the service with more technical information, such as whether the chat is web-browser-based or delivered through a specific platform, and include additional relevant technical specifications to clarify the mode of delivery- also, a screenshot could be added to the suppl. material.

Methods

Please clarify whether, in theory, young people could access both interventions (i.e., the synchronous chat service and the in-person/video counselling). And if you have assessed this- or can add information about it?

Consider revising or supplementing the terminology used for the two intervention types to align more closely with established terminology in e-mental health research. Specifically, you might include terms such as online synchronous chat counselling or online text-based support for the Live Chat service, and face-to-face/video counselling for the Jigsaw Brief Intervention. While the current terminology becomes clear upon reading, adopting more standard descriptors would make the distinction between the “text-based online” and the “visual/video or in-person” modalities more immediately evident. The term “Brief Intervention” alone may not sufficiently capture these differences.

Please also describe how suicidal ideation and risk were assessed within both interventions, including whether standardized screening or structured risk assessment protocols were used, and indicate how often SOPs for managing risk had to be implemented.

Results / Discussion

In the Discussion, please relate the findings to similar chat-based services in other countries. For example, consider referencing krisenchat (Germany), where the typical user profile (e.g., female, average age 17) may provide useful context for cross-country comparisons.

Additionally, please compare and discuss your findings regarding the prevalence of suicidal ideation or self-harm among users with results reported in other studies. Adding references to relevant international literature would strengthen the interpretation and external validity of the findings.

---

## [Reviewer Report]

Thank you for the opportunity to review this manuscript examining a live chat service designed for young people, with a valuable real-world comparison to in-person counselling.

The following feedback is intended to support the authors in refining their work, with specific, actionable recommendations organized by topic.

To be considered throughout the manuscript

To improve readability, I encourage the authors to review the consistency of abbreviations for the service names (e.g., the Irish Jigsaw Live Chat (JLC) and Jigsaw Brief Intervention (JBI)) throughout the manuscript. Consistent use of these abbreviations will enhance clarity for readers, reduce potential confusion, and ensure the manuscript maintains a professional and coherent presentation.

Abstract

The abstract section is clear and well-written. However, it would be beneficial to briefly describe the quantitative data analysis method within the methods section of the abstract. This provides readers with a clear overview of the analytical rigor and transparency of the study (e.g., “Data were analysed using descriptive statistics and logistic regression.”).

When reporting participant numbers, it is recommended to use ‘N’ before the figures and to specify from which services the participants were recruited (e.g., N=xxxx (JLC)) This practice enhances clarity and allows readers to quickly understand sample characteristics.

Introduction

The introduction is generally clear. Consider moving the description of the JLC to the intervention section. Placing detailed intervention descriptions in their dedicated section improves logical flow and helps readers distinguish between background information and methodological details.

Interventions and data collection

Clarify whether participants can log in to previously created accounts in JBC and if multiple accounts per participant are possible. This affects data integrity and the interpretation of engagement metrics.

Outline the account creation process, including any authentication or identification steps, and specify any baseline data collected at registration or first session. Example: “Participants create accounts using an email address; baseline demographic and mental health data with YP-CORE are collected during initial registration.”

Clarify if YP-CORE is used for JBC compared to JBI, noting any differences. Example: " The YP-CORE test is given to JBI participants upon admission, and in the JBC group, the young person completes it while waiting for the chat to begin.”

State the operating hours or availability times for both services. Service accessibility can impact participant engagement and outcomes.

If possible, create a distinct section outlining data collection processes, including what data were collected, tools/instruments used, and the process flow. This improves readability and allows for easier replication by other researchers.

Explain in more detail how reasons for attendance were collected (e.g., open-ended text fields) and specify timing in relation to registration at both JBC and JBI. Example: "At registration at JBC, participants answered the open-ended question: ‘What brought you to Jigsaw today?’

Service engagement characteristics.

Is it possible to include an analysis comparing the engagement profiles of JLC and JBI participants, as mentioned in the abstract? This would enrich the results. If it is not possible to perform the analysis, could it be considered in the discussion section?

Clearly outline which metrics (e.g., number of sessions, duration, dropout rates) were used to assess service engagement and define “routine data.” Transparency in metrics allows readers to interpret and compare findings accurately.

Evidence Gaps and the Current Study

Consider referencing the KidsHelpline report in this section.

Content analysis

Provide a detailed account of how qualitative responses were managed, including whether anonymized participant IDs were assigned and if software like NVivo was used. Assigning IDs ensures confidentiality and enables tracking responses across sessions, which is important for data integrity and longitudinal analysis.

The inductive content analysis and the development of the coding framework require a more detailed description. Elaborate on the process for developing the coding framework, referencing established methodologies (e.g., Campbell JL, Fonteyn ME, Gale NK, Kurasaki KS, MacPhail C). Example: “Coding was developed inductively following the approach described by Gale et al. (2013).”

Results

Clarification is needed regarding the source of information indicating that young people sought mental health support, did not report their reason, or supported another individual. Additionally, demographic characteristics are missing from JBI participants. If possible, add demographic data for JBI participants to Table 1. This facilitates direct comparison between groups.

The current presentation of Table 2 is unclear, particularly under the “YP-CORE clinical ranges” column. At present, only gender and age range are listed, and it is not specified whether additional variables are included or if these are the only variables considered. To improve clarity and ensure completeness, please indicate explicitly which variables are reported under the YP-CORE clinical ranges. If only gender and age range are included, state that no other variables are present. If other variables are missing, either add them or provide a note explaining their omission. This clarification will help avoid reader confusion and ensure the table is comprehensive and interpretable.

Explain how top risk behaviours and presenting issues were selected for reporting. Example: “The two highest-frequency risk behaviours (suicidal ideation, self-harm) and five most common presenting issues (anxiety, low mood, etc.) were included based on prevalence in the dataset.”

Update the table 3 title to indicate how information was obtained (e.g., “Clinician-Reported Issues”). This clarifies the origin of the data for readers.

Where appropriate, add p-values to tables to support statistical comparisons. This enhances the transparency and interpretability of reported differences.

It may also be beneficial to include the total duration of chat conversations. Reporting the overall length of chat sessions could provide further insight into participant engagement and the intensity of support provided. This additional metric would help contextualise the data and enhance the comprehensiveness of the results.

Supplementary material includes two table 3.

Reviewer’s Note on AI Assistance:

In preparing this review, I used AI Copilot to refine language, enhance clarity, and improve overall readability. Copilot also supported the structuring of the text, suggested improvements to coherence, and ensured consistency in terminology. These tools contributed to a more efficient writing process and the production of a clear and professionally presented review.

---

## [Editor Report]

Dear Authors 

We have now received reviewer comments on your manuscript. After reviewing their comments and suggestions we recommend minor revisions to be done to your submitted manuscript to help proceed. 

best regards

Siham

---

## [Editor Report]

Thanks for the revisions to the manuscript. We have reviewed the revisions and accept it for publication. 

Further details will follow

Regards

Siham